# Management of Bilateral Congenital and Juvenile Cataracts in a Low-Income Country: Patient Identification, Treatment Outcomes, and Follow Up

**DOI:** 10.3390/children11091064

**Published:** 2024-08-30

**Authors:** Broder Poschkamp, Serge Dinkulu, Thomas Stahnke, Clara Böckermann, Edith Mukwanseke, Christiane Paschke, Adrian Hopkins, Rainald Duerksen, Ellen Catrin Steinau, Andreas Stahl, Andreas Götz, Rudolf Guthoff, Ngoy-Janvier Kilangalanga

**Affiliations:** 1Department of Ophthalmology, University Medicine Greifswald, Ferdinand-Sauerbruch-Street, 17475 Greifswald, Germany; 2St. Joseph Hospital Kinshasa, J8MR+HCR, Kinshasa, Democratic Republic of the Congokilangalanga@cfoac.net (N.-J.K.); 3Institute for Biomedical Engineering, Rostock University Medical Center, Friedrich-Barnewitz-Street 4, 18119 Rostock, Germany; clara.boeckermann@uni-rostock.de (C.B.); andreas.goetz@med.uni-rostock.de (A.G.); rudolf.guthoff@med.uni-rostock.de (R.G.); 4Institute for Implant Technology and Biomaterials e.V., Friedrich-Barnewitz-Str. 4, 18119 Rostock-Warnemünde, Germany; 5German Committee for the Prevention of Blindness, Schulte-Marxloh-Str. 15, 47169 Duisburg, Germany; christianepaschkeconsulting@gmail.com; 6Adrian Hopkins Consulting, GAELF Secretariat, Liverpool School of Tropical Medicine, 76 Venture Court, Gravesend DA12 2AT, UK; adrianhopkinsconsulting@gmail.com; 7Christian Blind Mission (CBM), 14, Avenue Sergent Moke Concession Safricas, Ngaliema Kinshasa, Democratic Republic of the Congo; rainald.duerksen@cbm.org; 8Department of Gynaecology, University Medicine Greifswald, Ferdinand-Sauerbruch-Street, 17475 Greifswald, Germany; ellen.steinau@med.uni-greifswald.de

**Keywords:** cataract, pediatric, juvenile, congenital, bilateral cataract, international ophthalmology

## Abstract

Background: Childhood blindness remains a neglected issue in eye care within low-income countries, with congenital and juvenile cataracts being the most common treatable causes. This study aims to investigate the factors influencing the management of congenital and juvenile bilateral cataracts, provide data on general outcomes and postoperative findings, and discuss treatment in a low-income country context. Methods: In this prospective study, data from clinical care were continuously entered into a database, and one follow-up examination of a cohort of 91 patients with congenital and juvenile bilateral cataracts in Kinshasa, Democratic Republic of the Congo, was conducted. Multiple factors that affect the first clinical presentation, the clinical management, and outcome were investigated. Results: Most patients (88.5%) who received medical treatment were identified by cataract finders. A patient’s presentation time was independent of the parent’s education, social status, income, and sex of the child. The median age at first presentation was 5.8 years (ranging from 0 days to 17.3 years). The best visual acuity outcomes were achieved by patients who underwent surgery during early childhood. According to WHO grades and on an eye level basis, 51.1% (93 out of 182 eyes) showed improvement, while 34.6% (63 eyes) had unchanged results post-surgery. Among the cases without improvement, 76.2% were blind prior to surgery. Postoperative findings were observed in 56% of the patients, with secondary cataracts being the most common, affecting 26.37% of the operated eyes. Conclusions: Optimal management of childhood cataracts involves early identification, efficient hospital infrastructure, high-quality medical care, and long-term follow up. In sub-Saharan Africa, this approach must be adapted to the context of available resources, which differs significantly from those in high- and middle-income countries.

## 1. Introduction

Childhood blindness is a neglected problem in eye care in low-income countries [1]. The most common causes are cataracts, corneal scars, retinopathy of prematurity, glaucoma, retinal dystrophies, and retinoblastoma [2,3]. Congenital and juvenile cataracts are treatable causes of blindness in childhood. A congenital cataract is defined as a lens opacity that is present from birth or detected early in childhood (usually within the first year of life) [4]. Furthermore, cataract surgery is one of the most cost-efficient surgeries in healthcare [5].

Early visual deprivation during sensitive phases of visual development results in persistent visual impairment beyond surgery, because of amblyopia [6]. Thus, early identification and surgery in the early months of life as well as organized follow up are important components in the treatment of congenital cataracts. In low-income countries, the care of children with bilateral cataracts is challenging in terms of identification of children, resources, personnel and follow-up care.

This study aims to provide a comprehensive overview of congenital and juvenile cataract care in low-income countries, focusing on Kinshasa, Democratic Republic of the Congo (DRC). It explores influencing factors for patient identification, clinical management, and follow-up care. Additionally, it offers a detailed description of the outcomes of bilateral cataract surgery under conditions in a sub-Saharan country.

## 2. Materials and Methods

### 2.1. Study Description

A prospective study was conducted as part of a bilateral project between St. Joseph Hospital Kinshasa in the Democratic Republic of the Congo and the ophthalmology department of Rostock University Medical Center. From October 2018 to September 2022, 200 children with congenital or juvenile bilateral cataracts underwent surgery at St. Joseph Hospital. Rostock University Medical Center provided consultation and project coordination, while St. Joseph Hospital independently managed the medical care. Patient information was continuously pseudonymized and entered into a database developed by the University of Rostock. In September 2022, a follow-up re-examination was conducted on a stratified sample of 94 out of the 200 operated patients, selected based on age and sex. The inclusion criteria required an indication for juvenile or congenital bilateral cataract surgery and signed informed consent from the patient (if over 18) or a legal representative (if under 18) prior to surgery, conducted between October 2018 and September 2022.

Medical care included community-based screening of children using cataract finders, pre-examination, education, surgery, and follow up [1]. Cataract finders are trained to screen the population especially for cataracts using simple means (e.g., flashlight examination and visual acuity charts). The cataract finders organized screenings exclusively in Kinshasa, which were organized with local organizations, such as churches.

The cataract surgery criteria included patients with bilateral cataracts with a cataract greater or equal to 3 mm and reduced visual acuity. The surgical method used was the small-incision technique with primary IOL implantation and was carried out under general anesthesia using isoflurane and propofol. After capsulorhexis, most children’s lens materials were removed using manual irrigation and aspiration. For harder nuclear elements, a tunnel incision was made at the 12 o’clock position to extract them. A routine posterior capsulorhexis or capsulotomy was conducted, combined with an anterior vitrectomy using the Vitron unit (Geuder AG, Heidelberg, Germany). The intraocular lenses were then inserted through the tunnel incision. These lenses were placed into the capsular bag, or, if that was not possible, into the ciliary sulcus. Following the insertion of the lenses, viscoelastic material was removed, and the incisions were checked for watertightness. If needed, 10–0 nylon sutures were applied.

### 2.2. Follow-Up Examination

The follow-up examination was conducted at the office of St. Joseph Hospital in Kinshasa (DRC). Slit lamp examinations were carried out by specialized ophthalmologists with many years of experience. The examination included an evaluation of the anterior segment of the eye and a funduscopic examination under drug-induced mydriasis. If there were documented abnormalities where mydriasis should not be performed, such as a ciliary sulcus IOL, a funduscopic examination under miosis was conducted. Uncooperative children were re-invited for a short-term control.

Visual acuity was determined by use of LEA Symbols in 3 m [7]. For preschool children above 5 years of age, pre- and postoperative visual acuity assessment was carried out with Snellen Tumbling E or numbers. In cases where children did not respond to the LEA Symbols, a visual acuity examination using light perception (LP), hand movement (HM), counting fingers (CF) and no light perception (nLP) was determined. Visual acuity was classified according to WHO grades as follows: mild vision impairment is defined as visual acuity worse than 6/12 but equal to or better than 6/18. Moderate vision impairment is defined as visual acuity worse than 6/18 but equal to or better than 6/60. Severe vision impairment is defined as visual acuity worse than 6/60 but equal to or better than 3/60. Blindness is categorized as visual acuity worse than 3/60 [8]. For the analysis of visual acuity, all visual acuity values were first converted to logMAR. The logMAR values for counting fingers (CF = 1.98 logMAR), hand movement (HM = 2.28 logMAR), light perception (LP = 2.7 logMAR), and no light perception (nLP = 3.0 logMAR) were taken from the literature [9,10].

Three different hand-held automatic refractometers were used to determine their suitability for this clientele (1: Righton Retinomax Screen, Right Group, Tokyo, Japan; 2: Spot Vision Screener, Welch allyn, Skaneateles Falls, NY, USA; and 3: plusoptiX A12R, plusoptiX, Nuremberg, Germany); in all cases, a retinoscopy was performed. The examination of binocular functions was only carried out for orientation purposes in individual cases.

Photo documentation of the anterior segment of the eye with the slit lamp (iPhone 5, Apple, Cupertino, CA, USA) and measurement of intraocular pressure using a noncontact tonometer ic100 (iCare, Vantaa, Finland) were performed. The eye pressure was determined as the average value from six individual measurements. If elevated eye pressure (>21 mmHg) was detected, the mean value from three sets of six measurements was recorded. In addition, families were interviewed about their self-reported socioeconomic status with the assistance of interpreters fluent in French and Lingala (an English translation of the questionnaire is provided in the Appendix A).

### 2.3. Statistical Analysis

#### 2.3.1. Analysis Logic

Different time intervals were investigated for the analysis. The time intervals from birth to first medical presentation and the intervals from first presentation to surgery were examined. Finally, the results of the medical examinations, which describe the follow up, are presented.

In the first time interval, the influencing factors of presentation type (self-initiative, cataract finder), number of siblings, highest parent school education, and income were investigated. In the second time interval, parental income was examined. The analysis of the follow up included an analysis of the visual acuity as well as postoperative findings and intraocular pressure.

#### 2.3.2. Statistical Testing

Cleaning and formatting of the raw data were performed using Python 3.11 (Python Software Foundation, Wilmington, DE, USA) and Microsoft Excel (Microsoft Excel V 2211, Redmond, WA, USA). Statistical analysis and visualization were performed with Python 3.11 and Prism (GraphPad 9.4.1, USA).

When comparing two independent samples, a given normal distribution (Shapiro–Wilks test *p* ≥ 0.05), and homogeneous variances (Bartlett’s test for homogeneous variances *p* ≥ 0.05), an unpaired T-test was performed. For inhomogeneous variances, a Welch test was completed. If these conditions (homogeneous variances and normal distribution) were not met, a Mann–Whitney U test was applied.

In the case of a comparison between three or more independent samples, a one-factor analysis of variance (ANOVA I) was performed if normal distribution (Shapiro–Wilks test *p* ≥ 0.05) and homogeneous variances (Bartlett’s test for homogeneous variances *p* ≥ 0.05) were assumed, with the Tukey test serving as the post hoc test. For inhomogeneous variances, a Welch ANOVA with the Games–Howell post hoc test served as the statistical comparison. When these conditions were not given, a Kruskal–Wallis test was performed. Dunn’s test was used as the post hoc test in this case.

If not specified otherwise, the median and the interquartile range are always shown in the violin plot. A box plot contains the median and the interquartile. In the bar plot, mean ± SD is used.

### 2.4. Ethical Approval

The study was conducted according to the guidelines of the Declaration of Helsinki and approved by the Ethics Committee of the Ministry of Higher Education and Universities (DRC, Kinshasa, No.: ESP/CE/93B/2022), 13 July 2022.

## 3. Results

### 3.1. Patient Cohort

For the study, 94 children were medically examined. One patient declined scientific use of his data after examination, and three patients could not be matched with prior findings and were excluded. The mean age of the 91 patients at the time of examination was 8.2 years, with patients included from 1 to 19 years of age (Figure 1A). Of these, 64.8% were male and 35.2% were female (Figure 1B).

### 3.2. Time Intervals between Birth and First Presentation

The time intervals between birth and first presentation had a median of 5.8 years, and a range from 0 days to 17.3 years (Figure 2B). Documentation of the date of first presentation was available for 87 of the 91 children. Most children were detected by cataract finders (88.5%, Figure 2C). The children detected by cataract finders were younger on average in comparison with children who presented on their own initiative (6.74 years versus 8.03 years). However, this comparison was not significant (Mann–Whitney U test, *p* = 0.36, Figure 2D).

The number of siblings had no influence on time to recruitment (Kruskal–Wallis test, *p* = 0.19), with most patients having between one and four siblings (Figure 2E).

The highest level of education of the parents was most frequently given as an urban school education (approx. 83.3%). The comparison between the education levels was not significant (Kruskal–Wallis test, *p* = 0.66), noting that the group size differed substantially, and three parent couples refused to make a statement (Figure 2F).

The time intervals from birth to first presentation related to the sex of the patients were not significantly different (mean female: 5.9 ± 4.7 years, mean male: 7.5 years; Mann–Whitney U test: *p* = 0.18, Figure 2G).

Parental income did not seem to have an influence on the time interval (Kruskal–Wallis test, *p* = 0.94). The median time to first presentation varied from 4.67 years for destitute to 6.31 years for low income (Figure 2H).

### 3.3. Time Intervals between First Presentation and Surgery

The time intervals between presentation and first surgery ranged from 0 days to 977 days. The median was 105 days (Figure 3B). This result was independent of the parents’ income (Kruskal–Wallis test, *p* = 0.25, Figure 3C). However, it should be noted that the intervals were smaller for poor families (88.1 days versus 213.6 days for the mean of all other groups). The dates of presentation and first surgery were documented for 74 patients.

### 3.4. Surgery and Follow Up

#### 3.4.1. Visual Acuity

The best corrected visual acuity was considered before and after surgery at the follow-up date on an individual eye basis according to the WHO classification [8]. We defined the change from a worse to a better visual acuity level according to the WHO scale as an improvement. An improvement was observed for 93 eyes (51.1% of 182 eyes). For 63 eyes (34.6%), the result remained the same after surgery according to the WHO classification. Of these, 76.2% were blind (Figure 4B).

Of all 110 eyes that were blind before surgery, 62 eyes (56.4%) improved. Of the eyes that had severe visual impairment before surgery, 86.4% improved.

Preoperatively, the data for seven eyes of five patients were missing. Postoperatively, the data for 12 blind eyes (from eight patients) could not be collected. This was due to the young age and contradictory reactions. A repeated examination could not take place in this context of the consultation. Of all the eyes, seven eyes became worse. Of the three eyes with cataracts that had “no visual impairment” before surgery, according to the WHO grade, all worsened. This could be cataracts where lens opacities were only partially located in the visual axis and were symptomatic (Figure 4B).

When considering each eye individually, the median preoperative visual acuity improved from logMAR 2.7 (corresponding to light perception) to logMAR 0.9 (decimal visual acuity of 0.125) (Mann–Whitney U test, *p* < 0.001, Figure 4C).

For the patients who were operated on bilaterally in one procedure, the time interval between birth and surgery was plotted against the visual acuity after surgery in logMAR for each eye (Figure 5A). Linear regression was performed for this purpose. It should be added that no distinction could be made between congenital and juvenile cataracts.

The visual acuity outcome worsened continuously, with a later date of detection and surgery. The best visual acuity was achieved by patients who were operated on very early. The linear regression has an R^2^ of 0.28 (*p* = 0.0026).

The regression line shows that the general visual outcome only reaches a logMAR visual acuity of 1.0 from the age of about 15 years (Figure 5A). However, the individual values show a large scatter. A high age should not discourage cataract surgery.

When the outcome is considered on a patient basis, the vast majority of patients benefited from the treatment. Considered by WHO grades, the largest group before surgery was the one of bilaterally blind patients (n = 56 patients, Figure 5B). After surgery, 18 of these patients remained bilaterally blind according to WHO (Figure 5C). Among them, however, were visual improvements that are not represented by the WHO classification, such as light perception or the ability to count fingers.

#### 3.4.2. Findings and Management

Of 182 eyes, 80 eyes (43.9%) had normal postoperative results, and 102 eyes (56.0%) had 123 postoperative findings (Figure 6). The most common postoperative finding in 48 eyes (26.4%) was secondary cataracts, of which 81.25% required appropriate therapy. The remaining percentage did not show any opacification in the optical axis.

This was followed by findings involving the iris, with iris capture in 12 eyes, 9 eyes with synechiae, 7 eyes with pupil distortion and 3 other findings (iris atrophy, iris defect). In addition, there was an occurrence of microcornea (10 eyes) and microphthalmos (6 eyes), which seem to be associated with familial cataracts. Findings with the new intraocular lens (IOL) were dislocation or decentration (10 eyes), pigmentation (2 eyes), and small scratches (1 eye) and opacities (1 eye; Figure 6).

The follow-up findings necessitated therapy or therapy adjustments in approximately 60% of eyes. While every patient with an intraocular lens (IOL) required glasses, adjustments to the glasses were necessary in 21.4% of all eyes. Additionally, 20.3% of all eyes required management for secondary cataracts. The remaining patients required short-term follow up, surgery (IOL dislocation), or repeated retinoscopy. Short-term follow up was primarily required due to borderline elevated eye pressure or the inability to conduct thorough examinations in young children (Table 1).

#### 3.4.3. Postoperative Intraocular Pressure

Intraocular pressure was plotted per eye according to biological age (Figure 7A). A single measurement consisting of five individual measurements was taken. The average success rate for measuring intraocular pressure was 76.1% (Figure 7B). Reproducible eye pressures could be measured at an age of 2 years (success rate: 62.5%).

From a total of 140 examined eyes, a mean value of 14.62 mmHg with a standard deviation of 6.34 was obtained at an age of 1 to 19 years. Sixteen patients were recorded with an abnormal eye pressure of more than 21 mmHg and four patients had highly elevated values, with an eye pressure of more than 30 mmHg.

## 4. Discussion

### 4.1. Study Limitations and Study Conditions

The Democratic Republic of the Congo has an unevenly developed health care system and not everyone has access to it, so that many diseases are detected and treated too late. Furthermore, there is an extreme mismatch between the number of medical professionals and potential patients. Under these conditions, the regular and time-saving documentation of medical data is a challenge. To enable an analysis of the data, a database was specially developed. Nevertheless, it is a time-consuming task to fill it with standardized routine data. Collecting electronic data in a country where power outages occur several times an hour is challenging, leading to potential temporary data loss. When interpreting the results of this study, external factors such as power cuts (even during surgery), and the lack of medicines or medical materials must be considered. The study period fell within the COVID-19 pandemic period, with additional regulations and new priorities influencing the results of the study, which led to more complicated follow up and reduced treatment adherence.

### 4.2. Surgical Treatment of Congenital or Juvenile Cataracts

In this article, no distinction was made between congenital and juvenile cataracts [4], because, in this study setting, it was not possible to distinguish between the groups. In the meantime, however, procedures have been established that can distinguish between congenital and acquired cataracts by means of electrophysiological examination [11].

It is recommended to start the treatment of congenital cataracts within the first weeks to first months of life [12]. In high-income countries, depending on the cataract presentation, it is recommended to perform surgery on congenital bilateral cataracts in the first 6–8 weeks [12].

From our research, we observed that initial patient presentations were too late and delays in patient care were high. For congenital and acquired cataracts together, we found a median time to first presentation of 5.8 years. In India, times to first presentation were found to be 4.0 years for congenital and 8.3 years for acquired cataracts [13]. In Japan, an average age of 2.6 years was recorded for age at first presentation in a mixed patient group of congenital and acquired cataracts [14].

Based on several studies, it is recommended to use the manual small-incision technique in low-income countries [15,16]. It is less expensive, with a similarly low rate of complications, and is easier to learn [16,17]. The disadvantage using this technique is an increased rate of astigmatism [16].

If the patient is less than 6 months old, an IOL is not regularly implanted in high-income countries, and, in the case of aphakia, refractive deficiencies are compensated with contact lenses [12]. In a clinical randomized trial, no advantage or disadvantage was demonstrated for one-stage IOL implantation versus two-stage aphakia with vision correction and subsequent IOL implantation [18].

This two-surgery procedure assumes that the children are under continuous surveillance and medical care is always available. In addition, surgical risks are significantly lower, so that delayed IOL implantation is justifiable. In low-income countries, medical resources are limited (surgeons, anesthesiologists, consumables, and equipment) and the risk of surgery is higher (infections, power cuts during surgery, equipment failures) [19]. Under these conditions, we recommend early bilateral surgery (within the first few months of life) with primary IOL implantation, if possible, at first presentation (or a few days after) and after careful examination and indication.

Delayed surgery leads to a significant decrease in visual prognosis. At the same time, it was found in this study that surgery for children with significant cataracts and decreased visual acuity leads to an improvement in visual acuity, even in older children. In our study, three eyes with no visual impairment according to WHO scale were operated on. These eyes had a partial cataract. Furthermore, partial cataracts can lead to disturbing visual phenomena, reduction of the visual field, and differences between distance and near visual acuity. Each of these cases was the subject of a detailed discussion. Surgery for mild cataracts that impair vision should be strictly indicated, as there is a possibility that patients may experience poorer vision postoperatively.

### 4.3. Patient Identification and Date Documentation

Identifying childhood illnesses presents a challenge, as children rely on their parents to bring them to healthcare services. In this study, cataract finders successfully identified the majority of children with cataracts using visual acuity charts and torch examinations, facilitating their referral to medical care. The time it took for children to present for treatment was independent of their parents’ education, social status, income, or the children’s sex. However, a significant limitation of the current detection program is the absence of red reflex testing as a standard examination, which is particularly crucial for detecting suspected cataracts in newborns and preverbal children. Incorporating this test into screening programs would further enhance early detection efforts [20].

Cataract finders collaborate with local organizations, such as churches, to systematically access the population. There is a tendency in communities to want to protect children with congenital blindness, which means that these children are kept secret and often remain hidden from the public eye. The involvement of cataract finders has facilitated the identification of these children, who might otherwise have remained undetected.

Education and training of local staff who screen children should play a central role in local programs. Screening for cataracts can be conducted alongside screening for other childhood diseases. From an ophthalmological perspective, it could be combined with a red reflex examination or Brueckner fundus reflex test, a basic refraction check, and age-related visual acuity testing [21]. For instance, in India, refractive screening using the simple Folding Foropter, a paper-based self-refraction tool, has proven effective for older children [22].

The scope and capacity of screening should be closely coordinated with local partners and opportunities. In Kinshasa, for example, community-based screening is carried out with the help of church partners [23]. This collaborative approach ensures a structured and effective way to reach and screen the children in need.

The documentation of the dates of birth and screening, first presentation, surgery, and follow up have proven to be useful Key Performance Indicators for the evaluation of medical care for congenital and juvenile cataracts, in addition to visual acuity, as documented in our study. The times from birth to surgery have been published several times, and international comparisons can be made using this metric [13,14].

### 4.4. Complications in the Management of Pediatric Cataracts

#### 4.4.1. Surgery and Early Postoperative Findings

Complications during the surgery are lens luxation, capsular tear, iris trauma, hyphema, and posterior capsule rupture with vitreous loss [24,25]. Immediate postoperative complications are endophthalmitis, uveitis, cornea edema, increased intraocular pressure, and wound healing disorders [24]. Also, small-incision cataract surgery can lead to surgically induced astigmatism [26].

#### 4.4.2. Later Postoperative Findings

The follow up must consider that a high number of patients require further treatment. Thus, it was necessary to for a posterior capsulotomy, e.g., with a YAG-laser, which had to be performed in 20.3% of cases due to secondary cataracts. In order to reduce the number of secondary cataracts, a posterior capsulorhexis during surgery after the IOL implantation should be performed, but the possible complications should be considered and manageable. Even if a posterior capsulorhexis is performed, a secondary cataract can occur over the course of months. If the secondary cataract remains untreated and affects the visual axis, it can lead to stimulation deprivation amblyopia. Risk factors for visual axis opacification are young age at operation time, early IOL implantation, and ocular abnormalities. Therefore, visual axis opacification is the most common complication after cataract surgery in children and occurs in up to 40% of cases. In some cases, surgical membranectomy might be necessary [27,28,29].

Adjustment of glasses was necessary in 21.4% of cases. A frequent refraction check-up is necessary because the child’s eye is still growing and thus developing a myopic shift. Correction for “working distance” at near to be prescribed and checked at regular intervals. Especially for children under the age of two, IOL calculation is difficult, and predicted and actual refractive outcomes often differ [30]. To reduce the amblyopia risk and enhance visual rehabilitation, it is important to correct refractive errors as soon as possible [28,29]. In some cases, patients needed to be seen again to perform a retinoscopy, due to compliance issues. This was necessary for five eyes.

One important long-term complication is glaucoma following cataract surgery (GFCS). It can even occur years after surgery, so long-term follow up is necessary [31,32,33]. Patients at a young age at surgery time have a higher risk for developing GFCS [31,34]. Prescription of glaucoma medication was necessary for four eyes. Glaucoma management and surgery as well as regular follow up with eye pressure measuring is difficult to perform in low-income countries. For children, cyclophotocoagulation can be considered as a treatment option in a low-income setting [35].

Detailed preoperative amblyopia and strabismus data were not available for the study. The on-site investigation found that the concept of amblyopia and its treatment are difficult to communicate. As only children with bilateral cataracts were operated on, the children’s visual orientation improved significantly, which was seen as a success by their parents. The fact that the children’s vision could be better if early intervention including preoperative amblyopia treatment had been performed is often doubted by their parents. Good compliance as well as regular follow up and therapy adjustment are necessary for a good outcome, which is difficult to manage in a low-income country [29].

Some of our postoperative findings may be caused by an increased inflammatory response in children and a higher rate of fibrinous uveitis after cataract surgery [36,37]. These are, for example, pigment deposits on the IOL in 2 eyes, IOL decentration in 10 eyes, iris capture in 12 eyes, synechiae in 9 eyes, and pupil distortion in 7 eyes. These IOL-related complications can require a second operation, such as IOL exchange or re-positioning. In our study, this operation was recommended for seven eyes. Other reasons for secondary surgery can be glaucoma or visual axis opacification [37]. Another late complication that requires long-term follow up is retinal detachment, with a risk of 5.5% during the first 10 years [38].

Early cataract surgery is only the first step in treating pediatric cataracts. Children’s eyes are still developing and growing, which leads to changes in the refraction, and complications can occur years after surgery. This requires regular follow up, even years after surgery to prevent irreversible vision loss, e.g., caused by optic nerve damage due to GFCS [32,34,39]. Due to the long interval between first presentation of the cataracts and surgery, deprivation amblyopia was already present preoperatively and led to a reduced postoperative visual outcome, even after the best surgical results. Nevertheless, patients subjectively profited from minor objective improvements by regaining vision for daily life orientation.

In low-income countries, the attendance at follow-up examinations is often low. A study at the KCMC hospital in Moshi, Tanzania, showed that especially female children, preoperative blind children, and children of mothers without education and income, and with several children are least likely to attend follow-up examinations. Lack of information about the necessity of follow up was also a cause for poor attendance [33]. Further research is necessary to determine the influencing factors for poor follow up and to find solutions for better access to follow-up examinations.

We recommend a regular and long-term follow-up regimen that includes the following examinations: visual acuity testing using standardized tests, slit lamp examination, funduscopy, measurement of intraocular pressure, and retinoscopy, with adjustment of glasses prescription according to age. Additionally, amblyopia and strabismus therapy should be included to enhance visual rehabilitation and outcomes.

The recommended follow-up intervals are as follows. Patients should be hospitalized for 3 to 7 days postoperatively. Bandages should be removed within 1 day to mitigate the risk of amblyopia, and patients should be discharged with bifocal spectacles or two pairs of spectacles (one for distance and one for near vision). The first follow-up examination should occur after 1 week, followed by subsequent check-ups at 2 weeks, 1 month, 2 months, 3 months, 6 months, and then at 6-month intervals up to 2 years postoperatively. Thereafter, annual follow-up intervals are recommended.

In cases of unilateral cataract surgery, it is important to note that the second eye may also develop cataracts within months (in our study, we operated only on bilateral cataracts). Children are more susceptible to secondary cataracts and may develop recurrent secondary cataracts even after surgical membranectomy or YAG capsulotomy.

In our setting, social integration was facilitated with the assistance of cataract finders (ideally the same who identified the children), who ensured that the patients started or continued their education and attended school.

## 5. Conclusions

The comprehensive investigation into the management of bilateral congenital and juvenile cataracts in the Democratic Republic of the Congo has shed light on the critical aspects of pediatric ophthalmic care in low-income settings. The study’s findings reveal a significant delay in the median age at first presentation, underscoring the urgent need for early detection and intervention to optimize visual outcomes for affected children. The integration of community-based screening programs, utilizing cataract finders, can facilitate early identification and treatment. Despite these delays, the observed improvement in visual acuity post-surgery underscores the effectiveness of surgical interventions, even when performed at an older age. However, the prevalence of postoperative complications, particularly secondary cataracts, emphasizes the importance of diligent follow-up care to maintain the benefits of surgery and address complications timely. Future research should investigate factors that influence the follow up.

In conclusion, managing pediatric cataracts in low-income countries presents complex challenges but also opportunities for significant improvement in outcomes through targeted interventions. Early detection, timely surgical intervention, and comprehensive postoperative care are paramount. This study serves as a call to action for enhancing pediatric eye care in similar settings worldwide, leveraging the insights gained to refine strategies and foster collaborations aimed at eradicating childhood blindness.

## Figures and Tables

**Figure 1 children-11-01064-f001:**
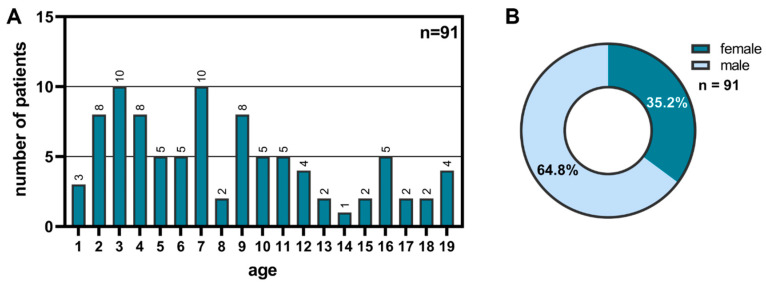
Representative sample of 91 patients from October 2018 to September 2022 (**A**) with the sex distribution of this sample (**B**).

**Figure 2 children-11-01064-f002:**
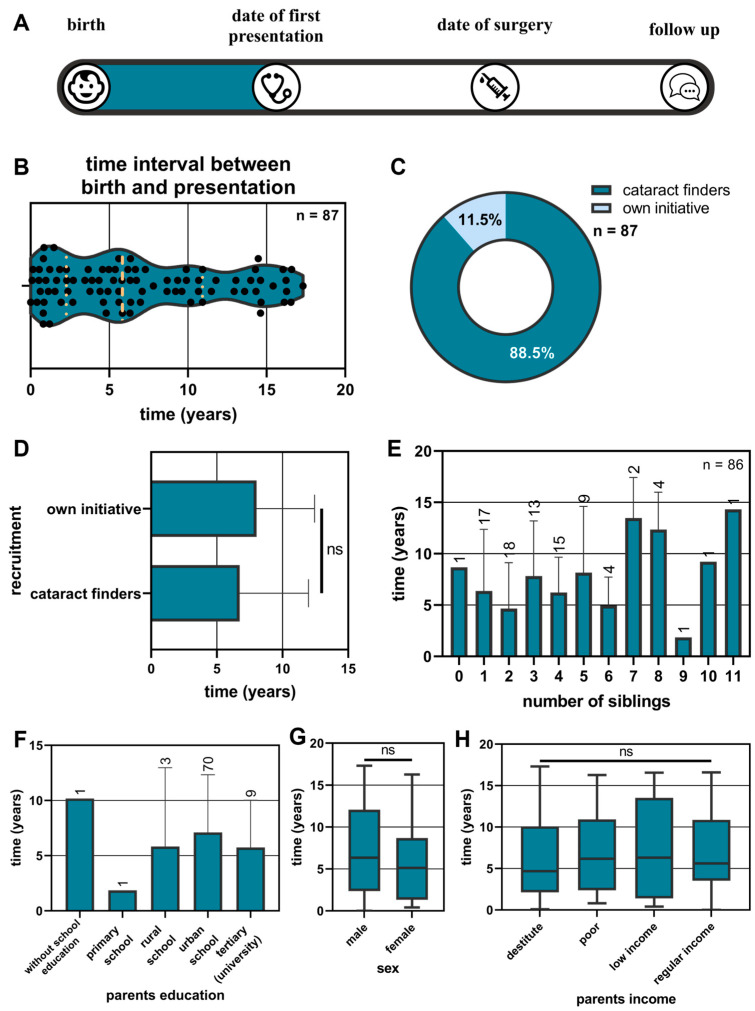
The results from the time intervals between birth and first presentation are presented (**A**). The time intervals between birth and first presentation are displayed using a violin plot. The median and interquartile range are marked with yellow dots (**B**). The recruitment types of patients (**C**) and the times based on recruitment are displayed (**D**). Different conditions for the time intervals between birth and first presentation are presented: the number of siblings (**E**), the education of the parents (**F**), the sex of the children (**G**), and the parents’ income (**H**).

**Figure 3 children-11-01064-f003:**
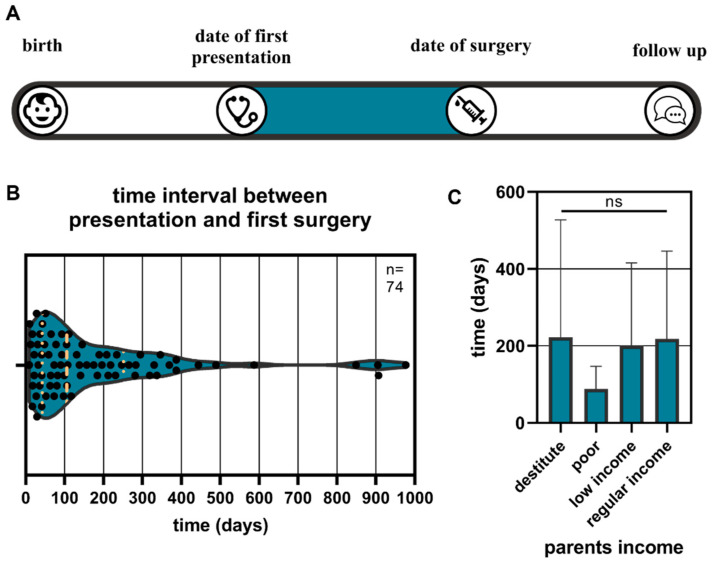
The time intervals between presentation and first surgery were investigated (**A**) and are displayed as a violin plot. The median and interquartile range are marked with yellow dots (**B**). The parents’ income in relation to the time intervals between first presentation and first surgery is presented (**C**).

**Figure 4 children-11-01064-f004:**
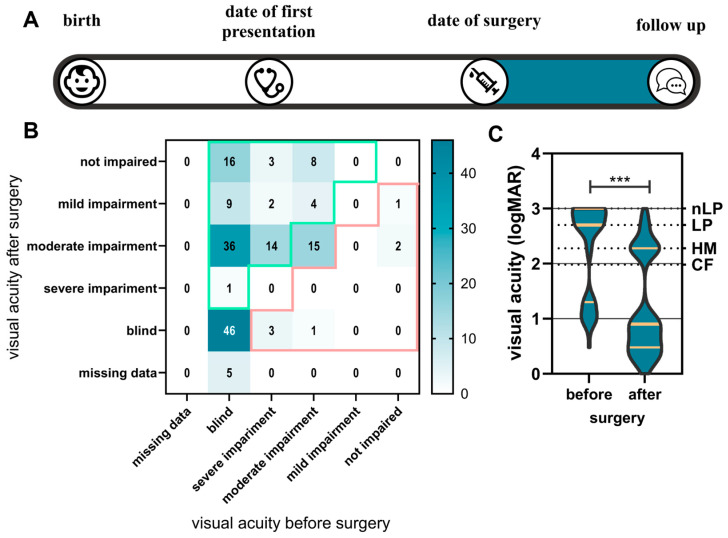
(**A**) The visual acuity per eye before and after surgery is displayed as a heatmap in WHO grades. (**B**) Green-bordered numbers represent an improvement and red-bordered numbers a decrease in visual acuity. The group size is indicated by numbers and blue coloring. (**C**) The visual acuity in logMAR before and after surgery is presented as a violin plot. The yellow coloring indicates the median, 25th, and 75th quartiles. ***: *p* ≤ 0.001, nLP: no light perception, LP: light perception, HM: hand movement, CF: counting fingers.

**Figure 5 children-11-01064-f005:**
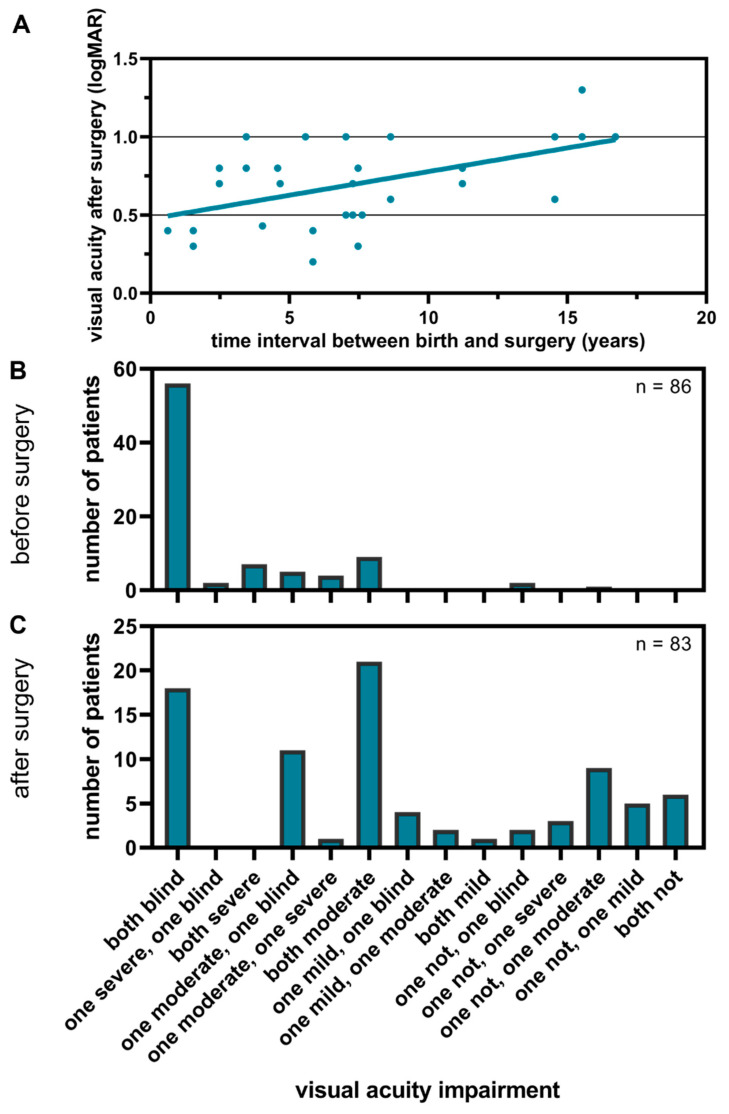
The visual acuity of each patient before and after surgery is displayed. (**A**) shows a scatter plot where each point represents the visual acuity after surgery (measured in logMAR), plotted against the time interval between birth and surgery for patients who underwent bilateral surgery in a single procedure. (**B**,**C**) present histograms illustrating the visual acuity distribution at the patient level before and after surgery, respectively. Please note that the y-axis scale differs between the two histograms.

**Figure 6 children-11-01064-f006:**
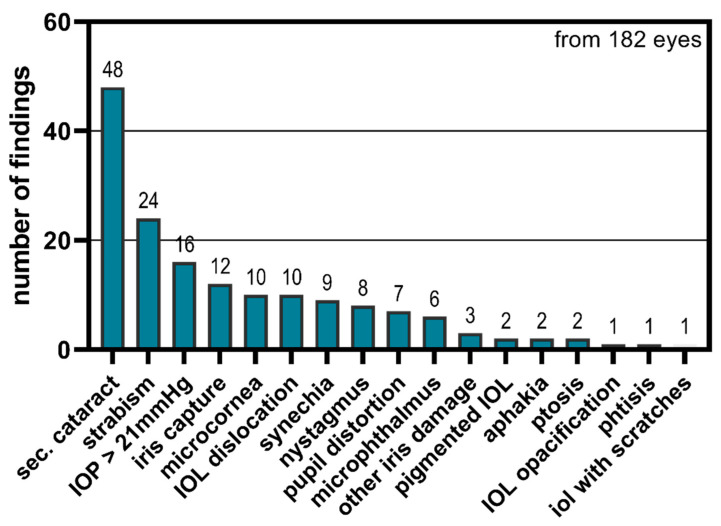
The findings from the follow up after congenital and juvenile cataract surgery are presented.

**Figure 7 children-11-01064-f007:**
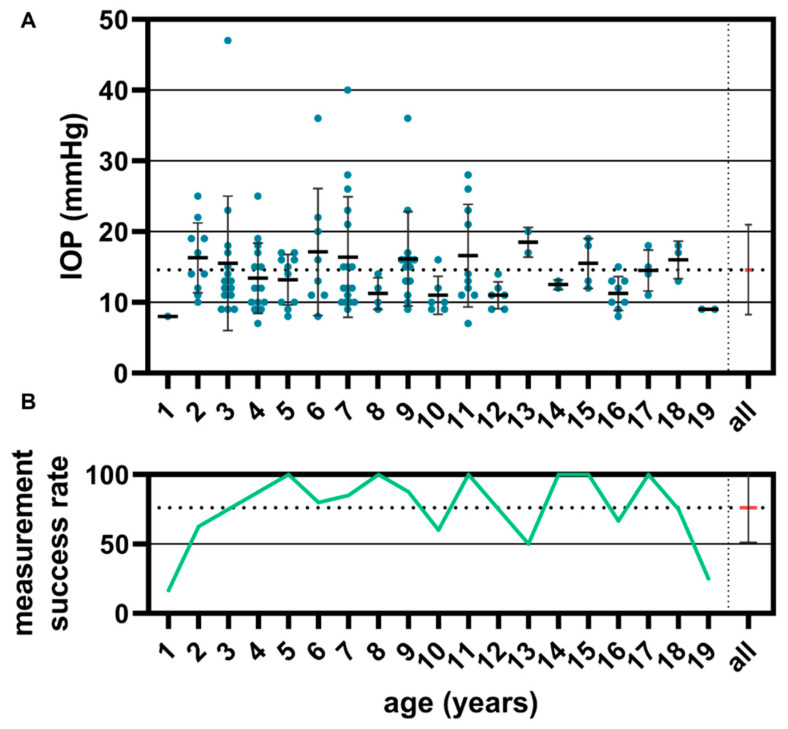
The intraocular pressure of eyes (n = 140) after cataract surgery (**A**), and the measurement success rate dependent on age (**B**) were investigated. The mean ± SD over all measurements have been marked with a red line.

**Table 1 children-11-01064-t001:** Management of findings after cataract surgery.

Management ^1^	Eyes	% (of All Eyes)
New glasses (after retinoscopy was done)	39	21.4
Posterior capsulotomy (YAG-laser)	37	20.3
Short-term follow up	15	8.2
Surgery	7	3.8
Repeated retinoscopy	5	2.7
Glaucoma medication	4	2.2

^1^ The management of squint and amblyopia was excluded.

## Data Availability

The data presented in this study are available on request from the corresponding author. The data are not publicly available due to privacy issues.

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
