# Peer review of "Management of Bilateral Congenital and Juvenile Cataracts in a Low-Income Country: Patient Identification, Treatment Outcomes, and Follow Up"

_children, 2024, doi:10.3390/children11091064_

Round 1
Reviewer 1 Report
Comments and Suggestions for Authors
It is a very broad sample on the results of bilateral congenital cataract surgery. Complications and the family environment are described. It is an excellent work on the difficulties of surgery in a low-income country.
1. Abstract. Improve the wording of this sentence: Of all patients 56% had postoperative findings, with the most common being secondary cataract (26.37% of operated eyes).
2. Introduction: This phrase corresponds more to the discussion than to the statement of the problem: Based on the data, recommendations are given for the management of bilateral congenital and juvenile cataract.
3. Material and methods: Definition of the concept of Childhood blindness. Is it 0.05 corrected visual acuity or less?
4. Describe more the surgical technique of cataract surgery.
5. Include cataract surgery criteria: greater than or equal to 3 mm, does it cause amblyopia or strabismus?.
6. Results: Materials and methods should include the visual acuity values ​​of visual impairment (light, moderate, severe) and blindness (3/60 or 1/60).
7. Results. Line 210: Expose Visual acuity mean preoperative (value) vs postoperative (Logmar value). The linear regression has an R² of 0.28 (p value significant or not).
8. The most important question of their work is whether: do children with bilateral congenital cataracts significantly improve LOGMAR visual acuity after surgery in their Kinshasa sample? Compare visual acuity pre LOGMAR vs visual acuity post LOGMAR. This question is partially answered in the descriptive improvement 56.1%, and in tables 5B and 5C a shift to values ​​of better visual acuity is observed. You need to demonstrate that there is an improvement and if possible satisfaction with the surgery (preoperatively and postoperatively this very dissatisfied, dissatisfied, satisfied or very satisfied with your vision).
9. Comment on discussion: the importance of Cataract finders in low-income countries
10. Explain in discussion why cataracts are operated on without visual impairment.
Author Response
Dear Reviewer,
We have also included our comments in a Word document for your convenience. You may use the document as needed.
Thank you.
__________________________________________________________________________________________________
Dear Reviewer,
Thank you for your detailed and thoughtful feedback on our manuscript. We appreciate your acknowledgment of our work and the valuable insights you have provided. We have carefully considered your comments and made the necessary revisions to address the concerns raised.
Comment 1: Abstract. Improve the wording of this sentence: Of all patients 56% had postoperative findings, with the most common being secondary cataract (26.37% of operated eyes).
Response 1: We have changed the wording to : “Postoperative findings were observed in 56% of the patients, with secondary cataract being the most common, affecting 26.37% of the operated eyes.”
Comment 2: Introduction: This phrase corresponds more to the discussion than to the statement of the problem: Based on the data, recommendations are given for the management of bilateral congenital and juvenile cataract.
Response 2: You are right! We have removed the sentence.
Comment 3: Material and methods: Definition of the concept of Childhood blindness. Is it 0.05 corrected visual acuity or less?
Response 3: We have used the visual acuity classification from the WHO. We have added the classification to the text. “Visual acuity was classified according to WHO grades as follows: Mild vision impairment is defined as visual acuity worse than 6/12 but equal to or better than 6/18. Moderate vision impairment is defined as visual acuity worse than 6/18 but equal to or better than 6/60. Severe vision impairment is defined as visual acuity worse than 6/60 but equal to or better than 3/60. Blindness is categorized by visual acuity worse than 3/60 [8].“
Comment 4: Describe more the surgical technique of cataract surgery.
Response 4: We have added a detailed description of the surgical technique: “The surgical method used was the small-incision technique and was carried out un-der general anesthesia using isoflurane and propofol. After performing capsulorhexis, most children's lens materials were removed using manual irrigation and aspiration. For harder nuclear elements, a tunnel incision was made at the 12 o'clock position to extract them. A routine posterior capsulorhexis or capsulotomy was conducted, combined with an anterior vitrectomy using the Vitron unit (Geuder AG, Heidelberg, Germany). The intraocular lenses were then inserted through the tunnel incision. These lenses were placed into the capsular bag, or if that wasn't possible, into the ciliary sulcus. Following the insertion of the lenses, viscoelastic material was removed, and the incisions were checked for w-ter-tightness. If needed, 10-0 nylon sutures were applied.”
Comment 5: Include cataract surgery criteria: greater than or equal to 3 mm, does it cause amblyopia or strabismus?.
Response 5: We only operated on bilateral cataracts if the cataract itself had a diameter of 3 mm or more and combined reduced visual acuity. We clarified it in the text.
Comment 6: Results: Materials and methods should include the visual acuity values of visual impairment (light, moderate, severe) and blindness (3/60 or 1/60).
Response 6: We have added it into the methods section.
Comment 7: Results. Line 210: Expose Visual acuity mean preoperative (value) vs postoperative (Logmar value). The linear regression has an R² of 0.28 (p value significant or not).
Response 7: Thank you for this remark! We added the p-value (p = 0.0026) which indicates a slope which is significant non-zero.
Comment 8: The most important question of their work is whether: do children with bilateral congenital cataracts significantly improve LOGMAR visual acuity after surgery in their Kinshasa sample? Compare visual acuity pre LOGMAR vs visual acuity post LOGMAR. This question is partially answered in the descriptive improvement 56.1%, and in tables 5B and 5C a shift to values of better visual acuity is observed. You need to demonstrate that there is an improvement and if possible satisfaction with the surgery (preoperatively and postoperatively this very dissatisfied, dissatisfied, satisfied or very satisfied with your vision).
Response 8: We have added a figure comparing preoperative and postoperative visual acuity in logMAR. The logMAR values for counting fingers (CF = 1.98 logMAR), hand movements (HM = 2.28 logMAR), light perception (LP = 2.7 logMAR), and no light perception (nLP = 3.0 logMAR) were obtained from the literature. Conversions and statistical calculations were performed in logMAR and subsequently converted back to Snellen (decimal) notation. Reference: Lange C, Feltgen N, Junker B, Schulze-Bonsel K, Bach M. Resolving the clinical acuity categories "hand motion" and "counting fingers" using the Freiburg Visual Acuity Test (FrACT). Graefes Arch Clin Exp Ophthalmol 2009; 247(1):137–42. Available under: https://link.springer.com/article/10.1007/s00417-008-0926-0. Thank you for your insightful comment regarding the importance satisfaction in Terms of Patient Reported Outcome Measures (PROMs). Unfortunately, we did not record the parents' and patients' satisfaction with the operation in this study. You are correct that PROMs are becoming increasingly significant in ophthalmology and have attained an important status. We plan to include them in future studies.
Comment 9. Comment on discussion: the importance of Cataract finders in low-income countries
Response 9: This study first revealed the importance of Cataract finders in Kinshasa. The work of the cataract finders cannot be valued highly enough! We have discussed the importance of this in more detail.
Comment 10: Explain in discussion why cataracts are operated on without visual impairment.
Response 10: We have described this topic in the discussion which was due to partial and symptomatic cataracts.

Reviewer 2 Report
Comments and Suggestions for Authors
Dear authors.
To begin, I'd like to congratulate the authors on their hard work and effort in carrying out their study, the design of which, regrettably, has not properly described in the Methods section. The goal was to explore influencing factors in the management of 91 children with congenital and/or juvenile bilateral cataracts in a low-income country. Furthermore, the study aimed to evaluate many factors influencing the first clinical presentation, clinical management, and outcome, with the ultimate goal of providing data on the general outcome, postoperative findings, and therapeutic alternatives. This study, in which the majority of the participants were identified by cataract finders, found that the presentation time was unaffected by the parents' level of education, social status, income, or gender. The median age at initial presentation was 5.8 years, and individuals who were operated on in their early childhood had the best visual acuity. It was later revealed that practically all patients, regardless of age, benefited from the procedure, with 56% experiencing postoperative anomalies, the most common of which being secondary cataract. Finally, the study indicated that the best therapy of childhood cataracts was a combination of early detection, efficient hospital systems, and appropriate medical care, all of which should be considered in the context of Sub-Saharan Africa.
I sincerely applaud your efforts on this research. Nonetheless, you presented a topic that had previously been extensively researched, not only regionally, but also globally. In terms of topic uniqueness and worldwide appeal in the literature, the subject appears to be fascinating, albeit somewhat boring; hence, I'm not convinced about its novelty in the literature. In addition, a number of concerns must be overcome in order for the task to be entirely literal. As a result, I believe this paper makes a limited contribution to the literature.
OTHER MEASURES
LANGUAGE
· The English language, in general, could be improved further because some sentences and phrases are difficult to understand, causing difficulty in interpreting and comprehending the study's overall results.
THE WHOLE MANUSCRIPT
· In terms of general manuscript analysis, this study looks to be excessively long, with numerous lengthy sentences that, in my opinion, distract the readers' concentration. With all due respect, I'm afraid I have to mention that it appears that the study was supposed to be a systematic review featuring a lot of stories about previously published works, which also indicates the study's lack of innovation. Therefore, please keep it succinct and to the point.
ABSTRACT
· I couldn't figure out why the name of the study location was designated as a keyword.
INTRODCUTION:
· Generally, the study design, as well as the location and time frame in which it was carried out, should be specified in the study methodology. I believe it would be more focused if the introduction section of the study was limited to outlining the overall notion of childhood cataracts, existing literature studies, and the gap that the study is about to address. Then, define the overall hypothesis driving the investigation, as well as the study's objectives.
· Page 2 line 58: It has been stated that ‘A study in a follow-up design…….’ I could not understand the term 'follow-up design study'. Were the authors referring to whether this was a prospective or retrospective study? Please double-check the issue here and elsewhere in the manuscript.
METHODS
· Page 2 line 79-80: It has been stated that ‘Inclusion criteria were signed informed consent from the patient or 79 a legal representative prior to surgery from October 2018 to September 2022.’ This appears to be inadequate in terms of clarity of inclusion requirements. Further, what characteristics were considered exclusionary. Furthermore, the phrase '….surgery from October 2018 to September 2022' should be removed from the sentence as it is already mentioned elsewhere in the manuscript.
· Page 2 line 81-84: ‘The mean age of the 91 patients at the time of examination was 8.2 years, with patients included from 1 to 19 years of age. Of these, 64.8 % were male and 35.2 % were female’. This information might be presented correctly in the study's results section.
· Page 3 line 91: It has been stated that ‘in case of abnormalities a funduscopic examination was performed.’ In fact, slit-lamp biomicroscopy of both the anterior and posterior segments is necessary for any pre-cataract surgery assessment. I do not believe that fundoscopy is only used when there are abnormalities, as the authors said in their manuscript. Please double-check and assure literature information as consistently firm as feasible for a rather commanded yet simple interpretation of the study as a whole.
RESULST
· Page 6 line 181-184: Please specify ''visual improvement. What criteria were used to assess these improvements?
· Page 6 line 189-190: This study raises various problems concerning data collection procedures. This has the potential to jeopardize the study's integrity and credibility overall.
· Page 6 line 191-194: It is understood that children with no visual impairment underwent surgery. The condition was then described in bewildering detail as being related with lens opacities that only affected the visual axis, rendering the children symptomatic. To be honest, this concept of surgery decision should be made completely plain, as well as clarifying the deterioration of postoperative outcomes rather than the better ones that should have been expected altogether.
· Page 7 line 204-205: ‘The visual acuity outcome worsens continuously with a later date of detection and surgery’. Do the authors mean ‘worsened’? Or purposely ‘worsens’? If the latter is true, it is obvious that there should be a reference to support the verdicts. Kindly clarify.
DISCUSSION
· Page 11 line 291: I couldn't locate anywhere that revealed how many children had both small-incision cataract surgery and secondary IOL implantation, or else. What are the authors' thoughts on this matter?
· Page 11 line 296: ‘Surgery for mild visual impairing cataract should be avoided.’ Would the authors detail the logic that led to this conclusion? Please!
· Page 12 line 327: Please kindly replace the phrase ‘ocular eye pressure’ with ‘intraocular pressure’
· Page 12 line 327-328: The authors mentions that ‘Also small-incision cataract surgery can lead to surgically induced astigmatism’. Did they attempt to gather data in this context? If yes, what was the mean/median postoperative astigmatism? Was the difference statistically significant as compared to pre-operative values? What steps were made to alleviate or perhaps decrease the complications that could have a significant impact on postoperative visual function, particularly in this group of patients who are still developing their vision?
· Page 12 line 353: Did any patients require glaucoma surgery? Or was the medication approach sufficient for this postoperative complication?
Comments on the Quality of English LanguageThe English language, in general, could be improved further because some sentences and phrases are difficult to understand, causing difficulty in interpreting and comprehending the study's overall results
Author Response
Dear Reviewer,
We have also included our comments in a Word document for your convenience. You may use the document as needed.
Thank you.
____________________________________________________________________________________________
Dear Reviewer,
Thank you for your detailed and thoughtful feedback on our manuscript. We appreciate your acknowledgment of our work and the valuable insights you have provided. We have carefully considered your comments and made the necessary revisions to address the concerns raised.
Comment 1: Language
Response1: We have revised the manuscript to enhance the clarity and readability of the English language. Specific sentences and phrases that were difficult to understand have been restructured for better comprehension.
Comment 2: In terms of general manuscript analysis, this study looks to be excessively long, with numerous lengthy sentences that, in my opinion, distract the readers' concentration. With all due respect, I'm afraid I have to mention that it appears that the study was supposed to be a systematic review featuring a lot of stories about previously published works, which also indicates the study's lack of innovation. Therefore, please keep it succinct and to the point.
Response 2: You are right that many different sources were cited in the manuscript and that it has a certain review character. It was intended to illustrate the difficult treatment situation faced in low-income countries by contrasting the desirable treatment with the treatment that is practically possible. We have taken your comments into account and reduced several parts of the manuscript.
Comment 3: ABSTRACT: I couldn't figure out why the name of the study location was designated as a keyword.
Response 3: We have re-evaluated the keywords and removed the specific mention of the study location as it is not essential.
Commet 4: Generally, the study design, as well as the location and time frame in which it was carried out, should be specified in the study methodology. I believe it would be more focused if the introduction section of the study was limited to outlining the overall notion of childhood cataracts, existing literature studies, and the gap that the study is about to address. Then, define the overall hypothesis driving the investigation, as well as the study's objectives.
Response 4: Thank you for your comments, we have incorporated them accordingly!
Comment 5: Page 2 line 58: It has been stated that ‘A study in a follow-up design…….’ I could not understand the term 'follow-up design study'. Were the authors referring to whether this was a prospective or retrospective study? Please double-check the issue here and elsewhere in the manuscript.
Response 5: We have clarified it.
Comment 6: Page 2 line 79-80: It has been stated that ‘Inclusion criteria were signed informed consent from the patient or 79 a legal representative prior to surgery from October 2018 to September 2022.’ This appears to be inadequate in terms of clarity of inclusion requirements. Further, what characteristics were considered exclusionary. Furthermore, the phrase '….surgery from October 2018 to September 2022' should be removed from the sentence as it is already mentioned elsewhere in the manuscript.
Response 6: You are right. Thank you for the remark, we have specified the inclusion criteria and removed the duplicate mention.
Comment 7: Page 2 line 81-84: ‘The mean age of the 91 patients at the time of examination was 8.2 years, with patients included from 1 to 19 years of age. Of these, 64.8 % were male and 35.2 % were female’. This information might be presented correctly in the study's results section.
Response 7: We have moved it to the results section
Comment 8: Page 3 line 91: It has been stated that ‘in case of abnormalities a funduscopic examination was performed.’ In fact, slit-lamp biomicroscopy of both the anterior and posterior segments is necessary for any pre-cataract surgery assessment. I do not believe that fundoscopy is only used when there are abnormalities, as the authors said in their manuscript. Please double-check and assure literature information as consistently firm as feasible for a rather commanded yet simple interpretation of the study as a whole.
Response 8: We must apologize for this mistake and thank you for pointing it out. It is not clear to us exactly how this could have been submitted like this after multiple proofreading. Of course, funduscopies in mydriasis were performed once before and several times after the operation for each patient. In the follow-up examination relevant for the study, funduscopy was also performed in mydriasis and only in some cases in miosis (in case of abnormalities such as ciliary sulcus iol to reduce the risk of iris capture).
Comment 9: Page 6 line 181-184: Please specify ''visual improvement. What criteria were used to assess these improvements?
Comment 9: We defined the change from a worse to a better visual acuity level according to the WHO scale as an improvement.
Comment 10 Page 6 line 189-190: This study raises various problems concerning data collection procedures. This has the potential to jeopardize the study's integrity and credibility overall.
Response 10: We have taken your point on serious consideration and discussed data collection problems in more detail. It is not easy to try to carry out electronic data collection in a country where power outages occur several times in an hour and where some temporary data is lost. We carried out the data collection with the utmost conscience. Unfortunately, in this case we were unable to provide better documentation.
Comment 11: Page 6 line 191-194: It is understood that children with no visual impairment underwent surgery. The condition was then described in bewildering detail as being related with lens opacities that only affected the visual axis, rendering the children symptomatic. To be honest, this concept of surgery decision should be made completely plain, as well as clarifying the deterioration of postoperative outcomes rather than the better ones that should have been expected altogether.
Response 11: We have provided a more detailed description of the surgical indication criteria. We also discussed these cases in more detail.
Comment 12: Page 7 line 204-205: ‘The visual acuity outcome worsens continuously with a later date of detection and surgery’. Do the authors mean ‘worsened’? Or purposely ‘worsens’? If the latter is true, it is obvious that there should be a reference to support the verdicts. Kindly clarify.
Response 12: We meant worsened. Thank you for your comment
Comment 13: Page 11 line 291: I couldn't locate anywhere that revealed how many children had both small-incision cataract surgery and secondary IOL implantation, or else. What are the authors' thoughts on this matter?
Response 13: In our study all patients underwent small incision cataract surgery with primary IOL implantation. We discussed the reasons for this in more detail.
Comment 14: Page 11 line 296: ‘Surgery for mild visual impairing cataract should be avoided.’ Would the authors detail the logic that led to this conclusion? Please!
Response 14: We have clarified it more: “Surgery for mild cataracts that impair vision should be strictly indicated, as there is a possibility that patients may experience poorer vision postoperatively.”
Comment 15: Page 12 line 327: Please kindly replace the phrase ‘ocular eye pressure’ with ‘intraocular pressure’
Response 15: Thank you for your
Comment 16 A: Page 12 line 327-328: The authors mentions that ‘Also small-incision cataract surgery can lead to surgically induced astigmatism’. Did they attempt to gather data in this context? If yes, what was the mean/median postoperative astigmatism? Was the difference statistically significant as compared to pre-operative values?
Response 16 A Unfortunately, it is very difficult to analyse the development of astigmatism due to small-incision cataract surgery in this model. This is due to the fact that paediatric cataracts are often very severe and therefore a significantly reduced fundus reflex can be perceived. This makes refraction determination with autorefractometers and via retinoscopy very difficult or even impossible preoperatively and would also mean additional stress for the patient, which is why this has not been intentionally investigated. We have quoted this from well-known studies on adults.
Comment 16 B: What steps were made to alleviate or perhaps decrease the complications that could have a significant impact on postoperative visual function, particularly in this group of patients who are still developing their vision?
Response 16 B: Many thanks for the advice. We have added precise information at the end to specify the recommended inspection intervals and the considerations to be taken into account. We included this section: “We recommend a regular and long-term follow-up regimen that includes the following examinations: visual acuity testing using standardized tests, slit-lamp examination, funduscopy, measurement of intraocular pressure, and retinoscopy with adjustment of glasses prescription according to age. Additionally, amblyopia and strabismus therapy should be included to enhance visual rehabilitation and outcomes.
The recommended follow-up intervals are as follows: Patients should be hospitalized for 3 to 7 days postoperatively. Bandages should be removed within 1 day to mitigate the risk of amblyopia, and patients should be discharged with bifocal spectacles or two pairs of spectacles (one for distance and one for near vision). The first follow-up examination should occur after one week, followed by subsequent check-ups at 2 weeks, 1 month, 2 months, 3 months, 6 months, and then at 6-month intervals up to 2 years postoperatively. Thereafter, annual follow-up intervals are recommended.
In cases of unilateral cataract surgery, it is important to note that the second eye may also develop cataracts within months (In our study, we operated only on bilateral cataracts.) Children are more susceptible to secondary cataracts and may develop multiple secondary cataracts even after surgical removal or YAG capsulotomy.
In our setting, social integration was facilitated with the assistance of cataract finders (the same who identified the children), who ensured that the patients started or continued their education and attended school.”
Comment 17: Page 12 line 353: Did any patients require glaucoma surgery? Or was the medication approach sufficient for this postoperative complication?
Response 17: We initially managed postoperative glaucoma with medication. However, this approach is challenging as many parents cannot afford the medication. Glaucoma surgery was not included in the program due to its high cost. We are currently seeking sufficient funding to purchase a cyclophotocoagulation device to better manage this complication.

Reviewer 3 Report
Comments and Suggestions for Authors
I would like to congratulate the authors on their work. This is a very interesting manuscript related to blindness prevention in children secondary to lens opacities.
Its always exciting to see how efforts have been made in a developing country to improve visual prognosis in pediatric cataracts, especially because early diagnosis is one of the most important factors to avoid permanent vision impairment in pediatric population.
As far as I know, this is the first paper that mention the advantages of a Cataract-case finder technician in pediatric population
The potential of this paper is very high and first of all, I would like to suggest a change in the title:
"Cataract identifiers technicians can improve pediatric lens opacity detection in a low-income African country"
I'd also recommend to analyse the same variables (both clinical: gender, diagnosis age, surgical time interval, preoperative and postoperative visual acuity using LogMar, surgical outcomes, etc, and demographic: parental income and education, number of siblings, etc) comparing two groups: cataract patients who sought medical attention brought by their parents and those who were recruited by cataract finders.
According to results, 85% of patients were identified by cataract finders technicians so this proportion will give very interesting results when both groups will be analysed and compared, and for sure, Cataract finders group may show many advantages in cataract detection, named: earlier detection and surgical treatment, better visual acuity, and possibly no gender differences among others.
The paper will turn from a descriptive study into a analytic one; With this approach, many of Figures and Tables can be replaced by fewer and new ones with variables comparison of both groups.
Pediatric cataracts visual prognosis is strongly related to early detection because lens opacities can affect normal eye development, so it is important to underline how Community Eye Care Workers could change the natural history of untreated cataracts in children. It is also important to discuss how visual rehabilitation has been made after surgical cataract removal, especially for those children under age of two y/o, and how complications could be related to age of surgery.
This cataract detection approach ("Cataract finders") would be replicated in other low-income countries both in African countries and also Asian and LatAm regions. So it will be important to describe how these technicians were trained and how they were working (city areas of screening, ophthalmological instruments used by them, etc).
In discussion, Red Reflex technique for lens opacities detection should be mentioned as a future improvement of the program (and can be easily performed by non-ophthalmology doctors and Community Eye Care Workers).
In case authors would change the sense of the manuscript, they should look for some references related to "cataract finders". Some of them are:
1) Devanesam Y, Rajaram SK. Engaging with communities to generate demand and improve the uptake of eye care. Community Eye Health. 2022;35:8-10
2) Razafinimpanana N, Nkumbe H, Courtright P, Lewallen S. Uptake of cataract surgery in Sava Region, Madagascar: role of cataract case finders in acceptance of cataract surgery. Int Ophthalmol. 2012;32:107-11.
3) Vanneste G. How eye workers can help newly blind people. Community Eye Health. 2003;16(45):5-6.
Author Response
Dear Reviewer,
We have also included our comments in a Word document for your convenience. You may use the document as needed.
Thank you.
_____________________________________________________________________
Dear Reviewer,
Thank you very much for your kind and encouraging comments on our manuscript. We are delighted to hear that you found our work on preventing blindness in children due to pediatric cataracts interesting and impactful.
We appreciate your recognition of the efforts made to improve visual prognosis in pediatric cataracts in a low-income country. Indeed, early diagnosis is crucial in avoiding permanent vision impairment in the pediatric population.
We are particularly pleased that you highlighted the role of cataract finders, as this is one of the major findings of our study. We were also surprised to see that these technicians managed to identify the majority of the patients, which underscores their effectiveness in this role.
Unfortunately, our study was not initially designed to specifically analyze the outcomes of children identified by cataract finders versus those brought by their parents. We have initiated a new investigation to explore and document the work of the cataract finders, and we plan to publish the results of this investigation later this year. The cataract finders work closely and effectively with local churches and communities.
While we appreciate your suggestion to change the title to "Cataract identifiers technicians can improve pediatric lens opacity detection in a low-income African country," we prefer to maintain the original title to reflect the broader scope of our study. Our intention was to present the outcomes of the patients and make comprehensive recommendations for low-income countries.
We also acknowledge your recommendation to analyze clinical and demographic variables by comparing the two groups. We have examined these variables and found no significant results for parental income, parental education, or the time interval between first presentation and surgery. If you believe our study would benefit from including these results, we can append them to the supplement or incorporate them into the results section.
Thank you for your heartfelt comments and constructive suggestions. They have provided valuable guidance for our future work.

Reviewer 4 Report
Comments and Suggestions for Authors
Abstract
Please the standard subtitles for Abstract presentation (Background, Methods, Results, Conclusions).
Please present details of the study design, information retrieved from the records, time frame of the chart review and patient presentation, etc.
Please present quantitative information on the study results (preoperative, operative and postoperative).
Manuscript
Introduction
"A study in a follow-up design was conducted on the management of 91 children with bilateral cataracts in Kinshasa, Democratic Republic of the Congo (DRC), to provide an exemplary holistic picture" 1. Please define (or rephrase) the "follow up" study design, is this prospective or retrospective? Or otherwise? 2. Please explain "exemplary". Recommendation is to avoid using abstract nonspecific terms.
Methods
"200 children with congenital or juvenile bilateral cataract were operated on" Please explain the basis of sample size calculation and recruitment of study participants (which community was screened? Which geographical location? Age group? etc)
"For the study, a representative sample (regarding age and sex) of 94 children was medically examined." Plesse explain the basis of choice of the "representative" sample (which age group or sex distribution was chosen)? What was the basis of exclusion of the remaining cases (to lower the count from 200 to 94 cases).
"Slit-lamp medical examinations were performed by ophthalmologists with many years of experience." Please mention the level of training of the ophthalmologists participating in the study (e.g. residents, specialists, etc)
"The examination included a focused examination of the anterior segment of the eye and in case of abnormalities a funduscopic examination was performed." 1. Define "focused" examination. 2. Was the examination performed in the office or under general anaesthesia? What about younger age groups that are uncooperative for office examination? 3. Fundus examination is routine (when media clarity permits) in any ophthalmic examination; please explain "in case of abnormalities".
"In cases where children did not respond to the LEA Symbols, a rough orientation as light perception (LP), hand movements (HM), and counting fingers (CF) was determined." Please explain the "rough" orientation. LP, HM and CF are determined by thorough examination. Please explain.
"The examination of binocular functions was only carried out for orientation purposes in individual cases." 1. Please explain "orientation" purposes. 2. Plesse explain the basis for selection of "individual" cases.
"The measurement was performed multiple times if a high intraocular pressure was found" IOP measurement for study purposes is routinely performed by multiple measurements and the the values averaged, not only for eyes with high IOP. Please explain.
"the families were interviewed about the socioeconomic situation" Please mention specifically the items investigated for the socioeconomic status (e.g. education level, occupation, annual income, etc).
Results
Please present information on the presentation in cases presenting by the care providers (e.g. White pupil, poor vision, ocular deviation, etc).
Visual acuity: "An improvement was observed for 93 eyes (51.1 %). For 63 eyes (34.6 %), the result remained the same after surgery according to the WHO classification. Of these, 76.2 % were blind" 1. Please mention the total number of eyes (not children) included in the study. 2. Please present information on the visual acuity of the study eyes in Snellen notation.
Since children were examined by ophthalmologists, please present information on the cataract morphology, on the IOP and fundus examination findings in study eyes.
Please present information on the surgical procedures performed, operative details, complications, intraocular lenses, etc.
"In remaining 40% the therapy for posterior cataract" Do the authors mean posterior capsule opacification?
Tanle 1: "Short term control surgery" Please define or clarify.
"3.3.3. Intraocular pressure" Were there values preoperatively or postoperatively?
Discussion
"Two main surgical techniques are commonly used in the treatment of cataracts: phacoemulsification and the manual small-incision technique [13]." This paragraph applies to adult cataract and is out of context to the current study topic. Please omit.
"4.4.1 Untreated cataract and surgery
The untreated mature cataract" The term "mature" cataract pertains to senile cataract and not childhood cataract. Please revise.
Is generally poorly written and not related to the current study findings. Please rewrite with focus on the current study findings.
Comments on the Quality of English LanguageEnglish language revision is mandatory.
Author Response
Dear Reviewer,
We have also included our comments in a Word document for your convenience. You may use the document as needed.
Thank you.
_____________________________________________________________________
Dear Reviewer,
Thank you for your detailed and thoughtful feedback on our manuscript. We appreciate your acknowledgment of our work and the valuable insights you have provided. We have carefully considered your comments and made the necessary revisions to address the concerns raised.
Abstract
Comment 1: Please the standard subtitles for Abstract presentation (Background, Methods, Results, Conclusions).
Response 1: Thank you for your advice. We have included it.
Comment 2: Please present details of the study design, information retrieved from the records, time frame of the chart review and patient presentation, etc.
Response 2: We have included the study design and also discussed further the information from records / database.
Comment 3: Please present quantitative information on the study results (preoperative, operative and postoperative).
Response 3: We have included the quantitative information on the study results. Thank you for your advice – this helped us to improve our abstract.
Introduction
Comment 4: "A study in a follow-up design was conducted on the management of 91 children with bilateral cataracts in Kinshasa, Democratic Republic of the Congo (DRC), to provide an exemplary holistic picture" 1. Please define (or rephrase) the "follow up" study design, is this prospective or retrospective? Or otherwise? 2. Please explain "exemplary". Recommendation is to avoid using abstract nonspecific terms.
Response 4: We have included the design and used more specific terms.
Methods
Comment 5 & 6:
-"200 children with congenital or juvenile bilateral cataract were operated on" Please explain the basis of sample size calculation and recruitment of study participants (which community was screened? Which geographical location? Age group? etc)
- "For the study, a representative sample (regarding age and sex) of 94 children was medically examined." Plesse explain the basis of choice of the "representative" sample (which age group or sex distribution was chosen)? What was the basis of exclusion of the remaining cases (to lower the count from 200 to 94 cases).
Response 5 & 6: We have specified the geographical location and provided a more detailed description of the screening process. Our project secured funding for surgeries on 500 children. After operating on 200 children, we decided to perform a follow-up on approximately half of these cases to assess outcomes and improvements. This sample size was chosen to provide a representative overview. To facilitate the follow-up, we covered travel costs, as parents were otherwise unable to bring their children for the necessary evaluations.
Comment 7: "Slit-lamp medical examinations were performed by ophthalmologists with many years of experience." Please mention the level of training of the ophthalmologists participating in the study (e.g. residents, specialists, etc)
Response 7: Thank you for your advice! We have included the level of training (specialists).
Comment 8: "The examination included a focused examination of the anterior segment of the eye and in case of abnormalities a funduscopic examination was performed." 1. Define "focused" examination. 2. Was the examination performed in the office or under general anaesthesia? What about younger age groups that are uncooperative for office examination? 3. Fundus examination is routine (when media clarity permits) in any ophthalmic examination; please explain "in case of abnormalities".
Response 8: We must apologize for this mistake and thank you for pointing it out. It is not clear to us exactly how this could have been submitted like this after multiple proofreading. Of course, funduscopies in mydriasis were performed once before and several times after the operation for each patient. In the follow-up examination relevant for the study, funduscopy was also performed in mydriasis and only in some cases in miosis (in case of abnormalities such as ciliary sulcus iol to reduce the risk of iris capture).
Our text now: “The follow-up examination was conducted at the office of St. Joseph Hospital in Kin-shasa (DRC). Slit lamp examinations were carried out by specialized ophthalmologists with many years of experience. The examination included an evaluation of the anterior segment of the eye and a funduscopic examination under drug-induced mydriasis. If there were documented abnormalities where mydriasis should not be performed, such as a cili-ary sulcus IOL, a funduscopic examination under miosis was conducted. Uncooperative children were re-invited for a short-term control.”
Comment 9: "In cases where children did not respond to the LEA Symbols, a rough orientation as light perception (LP), hand movements (HM), and counting fingers (CF) was determined." Please explain the "rough" orientation. LP, HM and CF are determined by thorough examination. Please explain.
Response 9: We have clarified it. It was determined by thorough examination. The choice of words was not well chosen. Thank you for your comment.
Comment 10: "The examination of binocular functions was only carried out for orientation purposes in individual cases." 1. Please explain "orientation" purposes. 2. Plesse explain the basis for selection of "individual" cases.
Response 10: In this study, we conducted examinations to detect strabismus. Unfortunately, comprehensive strabismus surgery and aftercare were not feasible due to limited personnel resources. Consequently, a detailed orthoptic examination was not performed. We are currently working intensively to train personnel to carry out these procedures in the future. We clarified it more in the text.
Comment11: "The measurement was performed multiple times if a high intraocular pressure was found" IOP measurement for study purposes is routinely performed by multiple measurements and the values averaged, not only for eyes with high IOP. Please explain.
Response 11: Thank you for your valuable advice. We specified our tonometry : “The eye pressure was determined as the average value from six individual measurements. If elevated eye pressure (> 21 mmHg) was detected, the mean value from three sets of six measurements was recorded.“
Comment 12: "the families were interviewed about the socioeconomic situation" Please mention specifically the items investigated for the socioeconomic status (e.g. education level, occupation, annual income, etc).
Response 12: We have added the Questionnaire in the supplement.
Results
Comment 13: Please present information on the presentation in cases presenting by the care providers (e.g. White pupil, poor vision, ocular deviation, etc).
Response 13: Unfortunately, we are unable to provide detailed information on this aspect, as the children were primarily identified by the cataract finders. These finders collaborated with churches and other local organizations to organize screenings. During these screenings, other health impairments (e.g., orthopedic or auditory issues) were also assessed. As a result, children referred for medical evaluation were always noted with 'suspected cataract'.
Comment 14: Visual acuity: "An improvement was observed for 93 eyes (51.1 %). For 63 eyes (34.6 %), the result remained the same after surgery according to the WHO classification. Of these, 76.2 % were blind" 1. Please mention the total number of eyes (not children) included in the study. 2. Please present information on the visual acuity of the study eyes in Snellen notation.
Response 15: 1.: We have added the total number of eyes. 2: We have added a figure comparing preoperative and postoperative visual acuity in logMAR. The logMAR values for counting fingers (CF = 1.98 logMAR), hand movements (HM = 2.28 logMAR), light perception (LP = 2.7 logMAR), and no light perception (nLP = 3.0 logMAR) were obtained from the literature. Conversions and statistical calculations were performed in logMAR and subsequently converted back to Snellen (decimal) notation. Reference: Lange C, Feltgen N, Junker B, Schulze-Bonsel K, Bach M. Resolving the clinical acuity categories "hand motion" and "counting fingers" using the Freiburg Visual Acuity Test (FrACT). Graefes Arch Clin Exp Ophthalmol 2009; 247(1):137–42. Available under: https://link.springer.com/article/10.1007/s00417-008-0926-0.
Comment 15: Since children were examined by ophthalmologists, please present information on the cataract morphology, on the IOP and fundus examination findings in study eyes.
Response 15: The documentation of cataract types (e.g., cortical, posterior, etc.) was inconsistent in our records. To conserve medical resources, this information was no longer recorded in the electronic database after an initial phase. Consequently, we are unable to provide detailed information on cataract types. We have described the results on eye pressure measurements in a separate section.
Comment 16: Please present information on the surgical procedures performed, operative details, complications, intraocular lenses, etc.
Response 16: We have described the surgical procedures in more detail in the study description.
Comment 17: "In remaining 40% the therapy for posterior cataract" Do the authors mean posterior capsule opacification?
Response 17: Thank you for your comment! We have improved and revised the section. We changed it to secondary cataract.
Comment 18: Table 1: "Short term control surgery" Please define or clarify.
Response 18: These are two separate rows. We meant short term follow up and surgery. We clarified it – thank you for your advice.
Comment 19: "3.3.3. Intraocular pressure" Were there values preoperatively or postoperatively?
Response 19: This was the postoperative intraocular pressure.
Discussion
Comment 20: "Two main surgical techniques are commonly used in the treatment of cataracts: phacoemulsification and the manual small-incision technique [13]." This paragraph applies to adult cataract and is out of context to the current study topic. Please omit.
Response 20: You are right. In this case, it distracts the reader.
Comment 21: "4.4.1 Untreated cataract and surgery
Response 21: We changed the discussion headings.
Comment 22 : The untreated mature cataract" The term "mature" cataract pertains to senile cataract and not childhood cataract. Please revise.
Response 22: We have changed the wording.
Comment 23: Is generally poorly written and not related to the current study findings. Please rewrite with focus on the current study findings.
Response 23: We have reevaluated the entire manuscript and placed a greater emphasis on our study findings.
We would like to extend our sincere gratitude. Your comments clearly reflect your deep interest in this topic and your extensive experience in the field. Your insights have significantly contributed to enhancing the quality of the manuscript.

Round 2
Reviewer 1 Report
Comments and Suggestions for Authors
The corrections requested by the reviewer have been made.
Author Response
We would like to express our sincere gratitude for your valuable comments. Your feedback has greatly enhanced the scientific quality of the manuscript.
Reviewer 2 Report
Comments and Suggestions for Authors
Dear authors.
As indicated in the prior review, I genuinely appreciate your work on this research, especially in endeavoring to undertake revision in a very focused and comprehensive manner. Despite your efforts, I'm afraid to inform you that this topic has already been extensively investigated, both regionally and globally. Frankly, this has nothing to do with the quality of revision on your work. As far as the topic uniqueness and universal appeal in the literature is concerned, the subject appears to be engaging, if somewhat boring. As a result, I'm not sure of its novelty in the literature. I really believe that this paper contributes little to the literature.
Comments on the Quality of English LanguageMinor editing of English language is required.
Author Response
Dear Reviewer,
Thank you very much for your thoughtful feedback and for acknowledging the efforts we’ve put into revising our manuscript. We genuinely appreciate your insights.
We understand your concerns regarding the novelty and global appeal of our study. However, we would like to emphasize a few key reasons why we believe our research is both important and relevant:
- Cataract Finders' Effectiveness: Our study found that cataract finders were highly effective in identifying children with cataracts and facilitating their access to necessary care. This is particularly important in Kinshasa, where early detection is critical for successful treatment outcomes. By sharing this finding, we hope to inspire similar strategies in other regions where early intervention can have a significant impact.
- Quantitative Reporting of Postoperative Findings: We’ve tried hard to provide a clear, quantitative picture of the postoperative challenges faced in this area. By doing so, we’re not just documenting the situation—we’re also helping to plan future projects more effectively. Understanding these challenges can lead to better integration of postoperative care in programs, which is essential for improving outcomes
- Local Relevance and Global Context: Kinshasa is one of the largest and fastest-growing cities in the world, projected to become the largest city by 2075. The healthcare challenges in such a rapidly expanding urban environment are immense, making our study highly relevant as we address the specific needs of this population. Additionally, it's important to note that not all African countries share the same conditions. Cultural, social, and economic factors can vary significantly, and our study reflects the unique context of Kinshasa, which may differ from other regions.
- Broader Impact for Sub-Saharan Africa: By detailing the situation in Kinshasa, we aim to provide insights that could be valuable for implementing similar programs in other parts of sub-Saharan Africa. While our findings are specific to Kinshasa, the lessons learned can inform and improve strategies in other regions, particularly those with similar urbanization challenges. We believe this study will contribute to the broader effort to enhance pediatric eye care across the continent.
- Relevance for International Comparisons: The detailed documentation of postoperative visual acuity and other time intervals also offers a basis for international comparisons. This data can help identify areas where practices can be improved or adapted to different contexts, providing a more comprehensive understanding of pediatric cataract care across various regions.
In conclusion, while the topic of cataracts has been studied extensively, our research offers unique insights into the specific challenges and successes within a rapidly growing urban area in a low-resource setting. We believe that our findings will not only improve care in Kinshasa but also provide valuable information for other regions facing similar challenges.
Thank you again for your valuable comments. We hope this explanation helps convey the significance of our work and its potential contribution to both local and global healthcare initiatives.
Reviewer 3 Report
Comments and Suggestions for Authors
Please mention in Discussion section that Red reflex examination as a future direction of this screening program.
Cataract finders and Community Eye Health workers should perform this examination: it is easy to perform, inexpensive and reliable, and most important: could be performed by non-ophthalmologist in newborns and toodlers
Mndeme, F.G., Mmbaga, B.T., Kim, M.J. et al. Red reflex examination in reproductive and child health clinics for early detection of paediatric cataract and ocular media disorders: cross-sectional diagnostic accuracy and feasibility studies from Kilimanjaro, Tanzania. Eye 35, 1347–1353 (2021). https://doi.org/10.1038/s41433-020-1019-5
Authors should mention lack of red reflex examination as a current detection program weakness, especially for suspecting cataracts in preverbal children.
Author Response
We would like to express our sincere gratitude for your valuable comments. Your feedback has greatly enhanced the scientific quality of the manuscript.
Comment 1: “Please mention in Discussion section that Red reflex examination as a future direction of this screening program.
Cataract finders and Community Eye Health workers should perform this examination: it is easy to perform, inexpensive and reliable, and most important: could be performed by non-ophthalmologist in newborns and toodlers”
Response 1: We have incorporated your feedback into our discussion: “Identifying childhood illnesses presents a challenge, as children rely on their parents to bring them to healthcare services. In this study, cataract finders successfully identified the majority of children with cataracts using visual acuity charts and torch examinations, facilitating their referral to medical care. The time it took for children to present for treatment was independent of their parents' education, social status, income, or the child's sex. However, a significant limitation of the current detection program is the absence of red reflex testing as a standard examination, which is particularly crucial for detecting suspected cataracts in newborns and preverbal children. Incorporating this test into screening programs would further enhance early detection efforts“
We also took you reference. Thank you for your valuable feedback – We incorporated it into our standard routine.
Reviewer 4 Report
Comments and Suggestions for Authors
Discussion
“The untreated cataract can lead to significant vision loss and even blindness. Intumescent or hyper-mature stages …… small-incision cataract surgery can lead to surgically induced astigmatism [30].” This information belongs to the Introduction section or can be omitted safely being unrelated to the current study findings.
“These are, for example, pigment deposits on the IOL in 2 eyes, IOL decentration in 10 eyes, iris capture in 12 eyes, synechiae in 9 eyes and pupil distortion in 7 eyes. These IOL-related complications can require a second operation such as IOL-exchange or reposition. In our study this operation was recommended for 7 eyes.” This information does not appear in the Results section. Please do not mention any information in the discussion not mentioned in the Results section
Comments on the Quality of English LanguageTrivial structure mistakes need correction.
Author Response
We would like to express our sincere gratitude for your valuable comments trough the review process. Your feedback has greatly enhanced the scientific quality of the manuscript.
Comment 1 : “The untreated cataract can lead to significant vision loss and even blindness. Intumescent or hyper-mature stages …… small-incision cataract surgery can lead to surgically induced astigmatism [30].” This information belongs to the Introduction section or can be omitted safely being unrelated to the current study findings.”
Response 1: We have omitted the part of the untreated cataract.
Comment 2: ““These are, for example, pigment deposits on the IOL in 2 eyes, IOL decentration in 10 eyes, iris capture in 12 eyes, synechiae in 9 eyes and pupil distortion in 7 eyes. These IOL-related complications can require a second operation such as IOL-exchange or reposition. In our study this operation was recommended for 7 eyes.” This information does not appear in the Results section. Please do not mention any information in the discussion not mentioned in the Results section. “
Response 2Thank you very much for your precise and insightful comments. We appreciate your attention to detail. The information you referred to was previously included in the figure but not fully described in the text. We have now incorporated it into the results section as follows: “This was followed by findings involving the iris, with iris capture in 12 eyes, 9 eyes with synechiae, 7 eyes with pupil distortion and 3 other findings (iris atrophy, iris defect). In addition, there was an occurrence of microcornea (10 eyes) and microphthalmos (6 eyes), which seem to be associated with familial cataracts. Findings of the new intraocular lens (IOL) were dislocation or decentration (10 eyes), pigmentation (2 eyes), and small scratches (1 eye) and opacities (1 eye; Figure 6).”